# Exploring Saudi Individuals’ Perspectives and Needs to Design a Hypertension Management Mobile Technology Solution: Qualitative Study

**DOI:** 10.3390/ijerph191912956

**Published:** 2022-10-10

**Authors:** Adel Alzahrani, Valerie Gay, Ryan Alturki

**Affiliations:** 1School of Electrical and Data Engineering, Faculty of Engineering and Information Technology, University of Technology Sydney, Sydney 2007, Australia; 2Department of Information Science, College of Computer and Information Systems, Umm Al-Qura University, Mecca 24382, Saudi Arabia

**Keywords:** hypertension, designing, mobile health applications, Saudi Arabia

## Abstract

Hypertension is a chronic condition caused by a poor lifestyle that affects patients’ lives. Adherence to self-management programs increases hypertension self-monitoring, and allows greater prevention and disease management. Patient compliance with hypertension self-management is low in general; therefore, mobile health applications (mHealth-Apps) are becoming a daily necessity and provide opportunities to improve the prevention and treatment of chronic diseases, including hypertension. This research aims to explore Saudi individuals’ perspectives and needs regarding designing a hypertension management mobile app to be used by hypertension patients to better manage their illnesses. Semi-structured interviews were conducted with 21 Saudi participants to explore their perspectives and views about the needs and requirements in designing a hypertension mobile technology solution, as well as usability and culture in the Saudi context. The study used NVivo to analyze data and divided the themes into four main themes: the app’s perceived health benefits, features and usability, suggestions for the app’s content, and security and privacy. The results showed that there are many suggestions for improvements in mobile health apps that developers should take into consideration when designing apps. The mobile health apps should include physical activity tracking, related diet information, and reminders, which are promising, and could increase adherence to healthy lifestyles and consequently improve the self-management of hypertension patients. Mobile health apps provide opportunities to improve hypertension patients’ self-management and self-monitoring. However, this study asserts that mobile health apps should not share users’ data, and that adequate privacy disclosures should be implemented.

## 1. Introduction

Chronic diseases, including diabetes, cardiovascular diseases, and hypertension, are among the main challenges facing healthcare systems worldwide and globally; they account for over 80% of all deaths [1]. In Saudi Arabia, chronic diseases are responsible for 73% of all deaths [2]. Furthermore, Saudi Arabia is fourth worldwide for individuals affected by hypertension [3], which is increasing in prevalence and affecting over one-fourth of the adult Saudi population [4]. These patients routinely need advice on nutrition, physical activity, and body weight [5]. The impact of this global epidemic can be reduced through appropriate control measures, including reductions in chronic disease risk factors, such as poor nutrition and physical inactivity. Mobile Health Applications (mHealth-Apps) that address risk factors of chronic diseases provide opportunities to improve the prevention and treatment of chronic diseases, including hypertension.

Over the last ten years, mHealth-Apps for health-related purposes have been increasingly used to support chronic disease management through mechanisms such as digital education, self-monitoring, and feedback [6,7]. The number of people accessing health-related information through mHealth-Apps on smartphones is fast growing. It is believed that more than five billion people around the world have mobile phones, and of these, more than half are smartphones [8]. Smartphone use has been proliferating in the Arab world, and in 2021, there were expected to be 45 million new smartphone owners [9,10]. Moreover, more than 300,000 mHealth apps are accessible worldwide, and that number is growing [11]. However, most mHealth-Apps lack scientific evidence, have not yet been tested for safety or usefulness, and some may even be hazardous [11,12].

MHealth-Apps are smartphone application programmes that provide health-related services. Nowadays, it is acknowledged that mHealth-Apps are essential in our lives and play a vital role in patients’ lives [13]. MHealth solutions are intended to motivate and persuade users to modify their behavior in order to assist them in accomplishing their health and wellness objectives. According to Birkhoff and Moriarty [14], app developers should create apps for varied patient populations, age groups, and literacy levels to improve patient engagement with the mobile technology solution and, ultimately, meet patients’ health requirements. Many patients with chronic conditions do not receive treatment that meets their needs; consequently, mHealth-Apps focus on advancing the delivery and organization of care for chronic condition patients and improving health outcomes. This paper aims to explore the requirements and needs of patients and medical experts to design a hypertension mobile app for Saudi patients.

## 2. Methods

### 2.1. Study Design

An exploratory qualitative study using semi-structured interviews was employed to understand the phenomena under investigation as experienced by the actors involved. This exploratory design allowed for the systematic collection, ordering, description, and interpretation [15] of narratives generated from each interview. After reviewing relevant literature on chronic diseases and the healthcare system in Saudi Arabia, a semi-structured interview guide was developed to explore the participants’ views about the demands and requirements of the users, as well as the usability rules, norms, and culture of mHealth-Apps. The interview type was semi-structured to allow the interview to be responsive to respondents’ answers and concepts emerging throughout the data collection process.

### 2.2. Study Setting and Participants

This study was conducted in Jeddah, Saudi Arabia. A total of 21 participants were identified through the researchers’ contacts and were interviewed. A snowballing technique was employed where initial respondents identified other relevant colleagues who were referred to the study. Participants were eligible to participate if they had worked or experience in hypertension management. Additionally, the inclusion criteria included patients who had had hypertension disease for at least one year, and were older than 18 years. All participants meeting these criteria were approached to participate in the study. All interviewees were contacted through email, with study information provided.

Interviewees were selected to contribute based on their expertise using purposive and snowballing sampling to identify relevant informants for different programs and management levels [16]. A sample of 21 participants were interviewed. The participants consisted of doctors and patients receiving treatments for hypertension at the Ministry of Health. Recruitment ceased after 21 interviews as it was evident that no new themes were emerging (saturation) [17].

The interviews took place at the workplace office, a meeting room in the public library, or a meeting room at the university of each of the interviewees (as per their preference). All interviewees interviewed were willing to participate and responded to the interview questions. There was no refusal of consent or drop-out during interviews. We ensured that there was no social desirability bias in this study, as there was no prior relationship between the interviewees and the interviewer.

### 2.3. Data Collection

Qualitative semi-structured interviews were undertaken to gather information about apps in relation to improving the management of hypertension with stakeholders, to attain the research aim. Open-ended interview questions were adopted to encourage a depth of understanding from each interviewee and to allow points of interest to be followed as they arose [15]. The interviewees’ interview guide included questions regarding their views on the demands and requirements of the users, as well as the usability rules, norms, and culture of mHealth-Apps. Interviews were conducted by (AZ) between March 2022 and June 2022. All interviews (*n* = 21) were conducted face-to-face at the workplace office of each of the interviewees, and the length of the interviews ranged from 20 to 30 min, with no repeat interviews conducted. As Arabic is the formal language in Saudi Arabia, all of the interviews were conducted in Arabic. Participant validation was achieved by returning transcripts to interviewees after completion of the interviews for comments or corrections. The transcripts were transcribed and translated from Arabic to English by a professional Arabic translator who spoke English as a second language to identify and resolve any issues with the concepts or expressions used. We ensured that there was no social desirability bias in this study, as there was no prior relationship between the interviewees and interviewer. The approval to conduct this study was granted by the ethics committee of the University of Technology Sydney, UTS HREC REF NO. ETH21-6571.

### 2.4. Data Analysis

The interviews were audio-recorded, transcribed, and coded into key themes. A thematic analysis framework was used for analyzing the data. This method consists of six steps: becoming familiar with the data, the search for themes, reviewing themes, defining and naming the themes, and finally, writing the research report [18]. The interview transcripts were re-read to ascertain that the identified themes formed a coherent pattern and captured the essence of the data. Transcripts were checked against audio files and entered into NVivo for management. NVivo software (Alfasoft, Gothenburg, Sweden) was used for organizing the data and facilitating the coding process. The first author (AZ) generated the initial data analysis and set of codes, and the interpretation was modified and refined in discussion with the second author (VG). A final agreement on the key findings and results was reached in consultation with both authors.

### 2.5. Results

A total of 21 participants were included in the study, and the mean age of the sample ranged from 41–59, and almost 23.8% of the sample belonged to the 30–40 age group. Gender composition was mostly male, as males represented nearly 66.7% and females represented nearly 33.3%. Out of the participants included, 52.4% had been diagnosed with hypertension for more than three years, and 33.3% had had the disease for 1 to 3 years, while 14.3% were newly diagnosed with hypertension and had had the disease for less than one year. More than half of the participants were educated up to the undergraduate level, at 57.14%. All the participants were using smartphones; however, only 14.3% were using Android, while 85.7 % were using IOS. The presence of characteristics among the participants is illustrated in Table 1.

This study grouped the findings according to the following emerging themes derived from the interviews into four categories. Themes and sub-themes were grouped according to whether they referred to: the perceived health benefit of the app, features and usability, suggestions for the app content, and security and privacy. Within the category of the perceived health benefits of the app, results were grouped by: physical activities, diet, and reminders. Within features and usability, results were grouped by: signing-up process, social communication and feedback, tracking, and self-monitoring.

## 3. Perceived Health Benefits of the App

The first theme deals with the perceived health benefit of the app. Many participants in this study were optimistic about the potential health benefits of the app, which could help people in general, and hypertension patients in particular, to adopt healthier lifestyles. Participants viewed apps which included physical activity tracking, related diet information, and reminders, as promising an increased adherence to a healthy lifestyle.

### 3.1. Physical Activities

Physical activity mobile apps have been reported to positively affect attitudes and perceived behavioral control. Engagement with physical activity mobile apps has been reported to be effective in behavior change interventions for hypertension patients. Participants shared that adaptation of the physical activities and availability of their daily steps positively influenced chronic disease patients’ encounters through better management of their diseases. Most participants were satisfied with apps that included physical activities and track their daily steps to manage their blood pressure and improve self-monitoring of hypertension. One participant stated, 

“*… I think apps that help me to accurately calculating my steps, walking distance time and calories will help me to manage my blood pressure (reading) and help me to monitor my hypertension*”.[P5]

### 3.2. Related Diet Information

An important way to stay on track with a healthy diet and healthy food intake is to schedule meals ahead of time and be familiar with the necessary nutrients of the meal. Eating healthily is crucial to reduce the risk of acquiring diseases, improve the management of diseases, and support physical well-being. A low-fat diet with lots of fiber and plenty of fruit and vegetables supports lowering blood pressure. Participants explained ways in which the diet inclusion in the apps made managing chronic disease patients more efficient and improved self-management. For example, one participant stated, 

“*My first goal from the health app is to control my nutrition in order to control my blood pressure, set goals, and reminder for healthy food*”.[P4]

### 3.3. Reminders

Participants identified several key benefits of reminders in the app in their response to the care of their diseases. These benefits can be broadly categorized as efficiency and useability. Participants emphasized multiple benefits of app reminders on efficiency, including accuracy of time. Interviewees highlighted the benefits of having reminders in the app data, which helps to remind them of the important activities to perform. Using a meal reminder app helps individuals stay on track with a healthy diet by reminding them to eat at the appropriate times. The apps will generate reminders and help track the time of daily food intake and physical activities. Reminders can be set up, and physical activities or food consumption can be tracked by day. One participant stated, 

“*… I need the app that keep reminding me to do my exercise and food suitable for BP*”.[P11]

Another participant stated, 

“*I like app that provide summary each day with steps, distance, calories, and remind me to set and achieve my goals*”.[P1]

## 4. Feature and Usability

The second theme captures the features and usability of the app. Therefore, this theme is made up of three sub-themes, as follows: signing-up process, social communication, and self-monitoring

### 4.1. Sign-Up Process

Participants reported that signing up for any mobile app should be a stress-free procedure, especially as it is the first interaction they will have with the application. Therefore, making registration easy, free, and quick in the apps is an important feature that needs to be taken into consideration. Most of the participants in this study preferred using free mobile apps. They believed that registering via an external account facilitates the registration of the apps, as external accounts can be used to speed up the registration procedure even more. A simple or two-click registration procedure may be achieved by allowing individuals to sign up using social media accounts like Facebook, Google, or Twitter. For example, one participant stated that, 

“*… the features of the app must include easy registration steps to be easy to use*”.[P4]

In addition, another participant stated that, 

“*easy registration should support social media like Facebook and Twitter because we can communicate via social media in Saudi, … and communicate with other users within the app*”.[P2]

Signing up using a patient’s ID, date of birth, and phone number are essential features of registration to the apps, and was reported by one participant who stated that, 

“*… I believe that apps should allow signing up through a patient’s ID, Date of Birth and Phone Number to allow doctors to access patients’ information and link to their electronic file*”.[P1]

### 4.2. Social Communication

The incorporation of mobile health apps and social media platforms into chronic disease prevention and management is expected to grow in practice as many individuals communicate online. Participants in this study reported that mobile health apps should incorporate social media to allow them to share goals and help them support each other, as well as share their experiences to improve the management of their diseases. The participants acknowledged that social media platforms’ interactive and participatory nature would allow for generating and sharing information, and increase engagement with other individuals who experience similar diseases and health professionals. Furthermore, participants in this study agreed that online engagement allows for discussion of their treatment plans in an easy way. One participant reported, 

“*I would prefer online engagement with my doctors to discuss my treatment plan when it is not necessary to travel to the clinic*”.[P7]

The willingness of individuals in this study to use social media for information and support suggests that tailored social media groups may provide a more acceptable, private way to seek such support.

### 4.3. Self-Monitoring

Most mobile health apps are developed to support patients’ self-monitoring of chronic diseases. Self-monitoring has an important role in improving clinical outcomes among patients with chronic diseases. Most participants acknowledged the benefits of using the apps to self-manage and monitor their diseases. Participants believed that mobile health apps could help them self-monitor their diseases and reach their goals. According to the participants, convenience was believed to be the main reason participants engaged with mobile health apps, which help them self-monitor and disease management. 

“*… Health apps should allow patients to set specific goals and send reminders about these goals*”.[P14]

On the other hand, the health professional who participated in this study believed that using mobile phones for routine self-monitoring and feedback is a cost-efficient strategy for self-management.

## 5. Suggestion for the App Content

The third theme focused on suggestions for the app’s content. It is critical to consider a user-centered approach for increasing the adherence to self-monitoring and the effectiveness of a mobile health, application-based intervention. Therefore, suggestions by potential future intervention users, and their expectations, should be involved during the design and development of a mobile health app for patients with chronic hypertension medical conditions. The participants of this study appreciated the ease of use of the apps, as well as immediate access to viewing their progress in charts and ability to share their progress with their family or health providers; one participant stated that, 

“*I would like to see my progress in graphs and hope to … share my progress with my family and my doctor*”.[P12]

In addition, participants prefer the apps to support Arabic; one participant said, 

“*I prefer the app to be in Arabic because in Saudi Arabia, the main language is Arabic, and I can speak Arabic only*”.[P18]

Additionally, the apps should provide instructions about choosing a workout or organizing healthy meals over a week. Participants agreed that the availability of this information could favorably affect users and allow them to have the appropriate level of preparedness required for performing physical activities and organizing their meals. Moreover, participants preferred easy registration to the apps through social media; as explained by one participant, 

“*If the registration through social media, it will make it very easy, I think*”.[P3]

## 6. Security and Privacy

The last theme dealt with security and privacy. Mobile health applications run on smartphones provide various ways to engage and empower users; thus, mobile health applications should promote security and privacy. Undoubtedly, the privacy of patients’ information is one of the most critical features of mobile health applications. Therefore, participants in this study believed it was crucial to take additional precautions when collecting and handling individuals’ health information. However, the majority of the participants were concerned about their privacy, and if apps shared their details; one stated that, 

“*I do not want the apps that share my home addresses and real-time location…. this is sensitive information… and I don’t want anyone to share it*”.[P17]

Another participant said, 

“*I am afraid if the apps share my details with other third parties, including Facebook, without taking permission from me*”.[P20]

All participants in our study agreed that it was critical to take a broader look at the risks of using health apps, as apps can pose dangers to physical integrity and users’ privacy.

## 7. Discussion

This study explored Saudi individuals’ perspectives and needs in relation to designing a hypertension management mobile app to be used by hypertension patients to manage their condition better. The present study aimed to gain an in-depth understanding of participants’ personal needs and suggestions for designing and developing mobile health apps for patients with hypertension. The outcomes of the user-centered developmental approach aimed to contribute knowledge on how to design and develop mobile health apps for hypertension patients. The most identified themes included: the perceived health benefits of the app, features and usability, suggestions for the app content, and security and privacy, and these will be discussed accordingly. The most popular categories of the mobile health applications, which participants expected to use daily, were physical activity tracking, related diet information, and reminders; promising an increased adherence to a healthy lifestyle.

The findings of this study highlight the health benefits of the mobile health app, which could help people in general, and hypertension patients in particular, to adopt healthier lifestyles. The study outlines that the apps that included physical activity tracking, related diet information, and reminders, promised increased adherence to a healthy lifestyle. This is consistent with the findings of another study by Wangler and Jansky [19], who claimed that using mobile applications (apps) to promote healthier lifestyles and aid in treatment appears to be highly viable in primary care. Therefore, using health apps has resulted in better behavioral changes [20]. Additionally, König et al. [21] found that smartphone health apps offered a potentially powerful new way to encourage people to adopt healthier lifestyles. Mobile health applications provide continuous monitoring of chronic diseases, higher quality of feedback and care, shorter hospital stays, and lower health care expenses to help patients achieve better health outcomes [20].

Moreover, mobile health apps have increased physical activity and improved nutritional habits [22]. The majority of apps for improving management of chronic diseases focus on modifying eating habits and physical activity levels as the primary targets, because these are the primary risk factors contributing to the development of these conditions [20,23]. Additionally, regular exercise can aid in the prevention and management of chronic illnesses such as hypertension, obesity, and type 2 diabetes [24,25,26].

This study revealed that the apps that included physical activity tracking, related diet information, and reminders are promising, and could increase adherence to a healthy lifestyle and consequently improve the self-management of hypertension patients. Likewise, Sardi et al. [27] acknowledged that, in mobile health apps, one of the most common types of notifications is a reminder. Reminders motivate users to use the app and make engagement with it more efficient, boosting their chances of meeting their targets [28]. In addition, Liu and Willoughby [29] supported that participants who received reminders enjoyed the messages and showed significantly higher levels of self-efficacy, knowledge of personal goals, motivation, and desire to use the app. Sohn et al. [30] observed that mobile health apps with tracking, tips, and reminders might help patients with chronic disease care.

Furthermore, high levels of inactivity, poor physical activity, and low physical fitness increase the risk of chronic illnesses, functional decline, and early mortality among older persons [31,32]. However, a well-balanced diet and frequent physical activity can help people maintain a healthy weight and lower their risk of chronic diseases [29]. According to König et al. [33], it has been shown that using nutrition apps can successfully change eating behaviors, as well as diet-related health risk factors. Health care professionals advise patients with diabetes or hypertension to use health apps to self-monitor their nutrition and physical activity [8].

Our study revealed that signing up for any mobile app should be a stress-free procedure, especially as it is the patient’s first interaction with the application. Therefore, an important feature that needs to be considered is making registration easy and quick. This is consistent with Xie et al. [34], who reported app developers needed to improve the usability of health applications to allow users with limited knowledge about health or mobile technology to use them efficiently. Moreover, usability must be considered from the start of system design to ensure the success of mobile health apps. According to [35,36], usability continues to be one of the most significant barriers to the adoption of mHealth technologies, because mHealth applications that are built with low quality are either difficult to use or are abused, which can lead to undesirable outcomes. The study of Vasiloglou et al. [8] indicated that an essential factor in choosing a health application was ease of use, followed by whether it was free to use.

Our study suggested that enabling registration via an external account could facilitate the registration of the apps. External accounts can speed up the registration procedure, allowing individuals to sign up using social media accounts like Facebook. Similarly, Burke-Garcia and Scally [37] argued that incorporating social media and mobile health in healthcare programs is beneficial due to the possible implications of mobile health apps in public health. One feature that sets most mHealth apps apart from other ways to track health information is sharing information with a user’s online social network [38]. Another study by Lee et al. [22] supported that most mobile apps track health status or behavior change, provide feedback, and deliver health-related information. Mobile health applications may contain goal setting, action planning, barrier identification, feedback, and social exchange functions [38]. Including some app features, such as self-monitoring and feedback, might be a motivator for continuous health app usage [33]. By adding this kind of feedback, applications continue to be very useful and powerful tools that help clinicians and improve patient safety [39].

The most frequently reported benefits of the mobile health apps included self-monitoring, which has an essential role in improving clinical outcomes among patients with chronic disease. This is consistent with the findings of a study by Waselewski et al. [40], which revealed that self-monitoring via mobile technologies can improve the management of chronic diseases in convenient and inexpensive ways. Similarly, Vaghefi and Tulu [28] identified that participants anticipated using the apps to monitor their progress towards their targets. The self-monitoring concept is connected to this idea, which states that when users can check their own performance, they are more likely to achieve their goals. This study acknowledges the benefits of using apps to self-manage and monitor their diseases. Similar findings were reported in a study by Thangada et al. [41], which argued that self-management of hypertension in mobile health applications could play a significant role in improving blood pressure control in hypertensive patients. Another study by Alturki [42] added that smartphone apps that allow patients to receive feedback and allow physicians to communicate with the patients’ could be valuable tools for improving blood pressure outcomes.

The existing evidence suggests that the content of mobile health apps needs to be perceived as personal and relevant to the patient’s own experience to increase treatment adherence. This study’s findings are consistent with Stephan et al. [43], who found that developing mobile health apps should be evidence-based, efficient, safe, and tailored to users and their needs. Another suggestion by participants in this study was the incorporation of social media into mobile health apps to enable easy registration.

Similarly, Alsalman et al. [44] reported that social communication is one of the most significant characteristics of mobile health apps that should be included in features. Social media and smartphone-based applications are now changing how people interact with healthcare and public health systems [45,46]. Social media can directly support disease management by creating online spaces where patients can interact with clinicians and share experiences with other patients [47]. Consequently, sharing health data among members of an online community seems also to correlate with better management of the disease they suffer from.

In order to successfully deliver services to the end user, it is essential to have deep familiarity with cultural norms and preferences [48]. Understanding the influence of culture on the acceptance of mHealth apps will help to develop and deliver new mHealth technology in the Arab world [48,49]. There has been relatively little study on the factors that influence physical activity in non-Western cultures, particularly the moderating effects of age and gender [50]. Alturki and Gay [51] stated that lack of exercise, due to the country’s climate, which encourages individuals to limit outdoor activities and stay indoors, is another critical issue in Saudi Arabia. One of the most important variables influencing an app’s performance is communicating with its users in a language they can comprehend. Arabic is a widely spoken language in the Middle East. Developers should consider local language use when designing applications that can improve the usefulness of apps and help spread their adoption by overcoming barriers like language [52]. Therefore, one of the most critical design elements that would have a significant impact on consumers’ use of mobile apps is the presence of the Arabic language [48]. There is still a lack of understanding of components of Saudi Arabian culture that must be considered when designing and implementing digital health programmes, particularly for chronic illness patients [53]. Saudi Arabia’s official language is Arabic. As a result, the app must be designed in Arabic, as this is the country’s primary language. Similarly, Alturki and Gay [54] highlighted that, for an app to be functional, it should adhere to the social and cultural standards of its users, such as being in the user’s native language and taking social traditions into account. Additionally, a significant number of participants in this study, more than 84%, were using IOS; therefore, the outcomes of this study support developing the mHealth-Apps on IOS.

Our findings indicate that app developers should not share users’ data and that adequate privacy disclosures should be implemented, as well as allowing users to make informed choices around the data. Similarly, Alturki et al. [55] emphasized the need to provide a detailed privacy policy in Arabic, and to inform users of the security procedures to protect their personal data. Consequently, patients should be informed about potential privacy risks before installation and use. Similarly, researchers conducting mHealth studies are increasingly concerned about patient data security and privacy [56]. Patient data is sensitive, and the vast amount of health data collected using mobile devices makes it hard to store data safely and securely, and encryption is often limited [57]. Mobile app development presents challenges related to patient privacy and data protection, high development costs, and the demand for easy-to-use designs [58]. Therefore, every developer working on an application must take the necessary precautions to protect an application and its content against nefarious efforts to compromise application data or user information [39].

Application companies often monitor users’ information, such as username, password, gender, age, and contact number, and sell some data to third parties [59,60]. Many people have smartphones but do not download mHealth apps because they are either not interested, think the apps are too expensive, or worry about their data being collected [61,62]. Users are concerned about the security risks of mobile apps and that apps may contain harmful software that follows their activity, steals sensitive data, and conducts unwanted phone calls [63,64]. Therefore, it is crucial to use the correct tools to examine mHealth apps’ privacy and security.

## 8. Implications for Practice and Future Research

The current study’s findings shed light on understanding the variety of user experiences and expectations of potential future intervention users. These can help designers create mobile health apps that encourage self-monitoring to improve self-management of hypertension. It provides practical insights into the challenges and needs of patients with hypertension, highlighting key areas to be addressed by policymakers and stakeholders to develop and design mobile health apps to improve the management of hypertension. Future studies evaluating the needs of hypertension patients should involve a quantitative approach with large samples of participants for a deeper understanding of the phenomenon, and to effectively explore mobile health app features that could help improve the management of hypertension and other chronic conditions.

## 9. Limitations and Strengths

Limitations of this study include conveniently drawn samples from the sample size. Nevertheless, the sample size was sufficient to identify the requirements and needs of patients and medical experts to design a hypertension mobile app for Saudi patients. As this research implemented a qualitative study design, we could not make causality assumptions, which is a limitation when interpreting the findings. Additionally, qualitative research is not statistically representative. However, it is much more appropriate for our research as it seeks to understand individuals’ perspectives and needs in relation to designing a hypertension management mobile app to be used by hypertension patients to manage their condition better. To the authors’ best knowledge, this study is the first of its kind to gain an in-depth understanding of participants’ personal needs and suggestions for designing and developing mobile health apps for patients with hypertension in Saudi Arabia.

## 10. Conclusions

The semi-structured interviews revealed the expectations of how individuals use health monitoring applications. Understanding the variety of user experiences and expectations of potential future interventions, users can help designers to create health apps that encourage self-monitoring perseverance that result in improved self-management of hypertension. The mobile health apps should include physical activity tracking, related diet information, and reminders, which promise an increased adherence to healthy lifestyles and consequently improve the self-management of hypertension patients. Mobile health apps have opportunities to improve hypertension patients’ self-management and self-monitoring. However, this study asserts that mobile health apps should not share users’ data, and adequate privacy disclosures should be implemented

Notably, the specification of frequencies in reported themes should provide a notion of their putative relevance. It might be essential to conduct online and face-to-face focus groups to gain more insights for future studies. Future studies should aim to better understand the perspective of other stakeholders in order to increase the uptake, adherence, and effectiveness of mobile health apps. Our findings provided important insights into the challenges and needs of patients with hypertension, highlighting key areas to be addressed by policymakers and stakeholders to develop and design mobile health apps to improve the management of hypertension. The patient-centered suggestions for the app content and the recommendations derived from this study may also be relevant for future mobile health app interventions; designed for patients with other chronic conditions, such as diabetes. These findings are instrumental in helping policymakers and stakeholders design and develop suitable mobile apps for hypertension patients’ self-monitoring and education, and consequently, improved care.

## Figures and Tables

**Table 1 ijerph-19-12956-t001:** Presence of characteristics among the participants *n* = 21.

User	Gender	Age Group	Time since Diagnosed with Hypertension (Years)	Education Level	Smart Phone Users	Type of Smart Phone
1	Male	>60	>3	Undergraduate degree	Yes	iPhone
2	Female	41–59	>3	Undergraduate degree	Yes	iPhone
3	Male	41–59	1–3	Undergraduate degree	Yes	iPhone
4	Male	>60	>3	Postgraduate degree	Yes	iPhone
5	Male	18–29	1–3	High school	Yes	iPhone
6	Female	41–59	>3	Diploma	Yes	iPhone
7	Female	30–40	>3	Undergraduate degree	Yes	iPhone
8	Male	>60	>3	Undergraduate degree	Yes	iPhone
9	Male	18–29	1–3	Undergraduate degree	Yes	Android
10	Male	41–59	>3	Postgraduate degree	Yes	iPhone
11	Female	41–59	<1	Undergraduate degree	Yes	iPhone
12	Male	30–40	1–3	Undergraduate degree	Yes	iPhone
13	Male	41–59	1–3	Undergraduate degree	Yes	iPhone
14	Male	18–29	>3	Diploma	Yes	iPhone
15	Female	41–59	>3	Undergraduate degree	Yes	iPhone
16	Male	30–40	<1	Diploma	Yes	Android
17	Female	41–59	1–3	Undergraduate degree	Yes	iPhone
18	Male	18–29	<1	High school	Yes	iPhone
19	Male	>60	>3	Undergraduate degree	Yes	Android
20	Female	30–40	1–3	Postgraduate degree	Yes	iPhone
21	Male	30–40	>3	Diploma	Yes	iPhone

## Data Availability

The datasets generated and/or analysed during the current study are available from the corresponding author on reasonable request.

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
