# Peer review of "Exploring Saudi Individuals’ Perspectives and Needs to Design a Hypertension Management Mobile Technology Solution: Qualitative Study"

_ijerph, 2022, doi:10.3390/ijerph191912956_

Round 1

Reviewer 1 Report

Thank you for the opportunity to review this paper. The authors have aimed to explore the requirements and needs of patients and medical experts to design a hypertension mobile app for Saudi patients. It is very interesting that the authors focused on development of a mobile app. It is a very relevant topic in the current world with the increasing use of smartphones and constant development of mobile apps. It was really nice that the authors used a qualitative design to understand the needs and preferences of individuals which will help with development of this app. Moreover, the world is seeing a constant rise in chronic diseases such as hypertension and such an app might even help individuals who are at risk for hypertension. The app can help those at risk to modify some of their lifestyle factors by increasing physical activity and practicing a healthy diet and monitor their progress.

Future studies conducted after development of such as app can even consider a mixed methods design where they analyze app data but the same time identify themes using focused groups.

Overall, the authors have done a nice job of presenting their findings. Several interesting themes have been captured.

A few minor comments:

·       Have other studies focused on mobile apps been done using a population in Saudi Arabia to examine other chronic diseases such as diabetes or CVD?

·       Methods: was there a cut-off for age considered?

·       In which language were the interviews conducted? If so, were the transcripts translated to English?

·       Please mention in the methods about who transcribed the interviews?

·       Was any information collected on whether people had a preference towards only using free mobile apps? Several apps come with a feature of becoming a paid member as that offers additional benefits. It would be interesting to know if similar information was captured in the interviews. If the app is not free, then the tendency to purchase a plan will decrease.

Author Response

We greatly appreciate the reviewers’ enthusiasm about the manuscript and the quality comments offered by the reviewers. The suggestions offered by the reviewers have been immensely helpful, and we also appreciate your insightful comments on revising the paper. We have included the reviewers’ comments in the following table and responded to them individually, describing the changes we have made. Also, all changes are highlighted and have been incorporated into the manuscript, and the revised manuscript is attached.

Author's Reply to the Review Report (Reviewer 1)

Thank you for the opportunity to review this paper. The authors have aimed to explore the requirements and needs of patients and medical experts to design a hypertension mobile app for Saudi patients. It is very interesting that the authors focused on development of a mobile app. It is a very relevant topic in the current world with the increasing use of smartphones and constant development of mobile apps. It was really nice that the authors used a qualitative design to understand the needs and preferences of individuals which will help with development of this app. Moreover, the world is seeing a constant rise in chronic diseases such as hypertension and such an app might even help individuals who are at risk for hypertension. The app can help those at risk to modify some of their lifestyle factors by increasing physical activity and practicing a healthy diet and monitor their progress.

Future studies conducted after development of such as app can even consider a mixed methods design where they analyze app data but the same time identify themes using focused groups.

Overall, the authors have done a nice job of presenting their findings. Several interesting themes have been captured.

A few minor comments:

  • Have other studies focused on mobile apps been done using a population in Saudi Arabia to examine other chronic diseases such as diabetes or CVD?

There are some studies on diabetes, but there is no such study on Saudi population with hypertension. To the authors' best knowledge, this study is the first of its kind to gain an in-depth understanding of participants' personal needs and suggestions for designing and developing Mobile health apps for patients with hypertension in Saudi Arabia.

  • Methods: was there a cut-off for age considered?

There was no  cut-off for age considered in this study, as highlighted in the method section in the inclusion criteria.

  • In which language were the interviews conducted? If so, were the transcripts translated to English?

The interviews were conducted in Arabic language, and the transcripts were translated to English Language.  This comment has been addressed and incorporated in the manuscript under the method section, please see the yellow highlight on page (3)

  • Please mention in the methods about who transcribed the interviews?

Thanks for your comment, this has been addressed under the method section please see page (3)

As Arabic is the formal language in Saudi Arabia, all of interviews were conducted in Arabic language. Participant's validation was achieved by returning transcripts to interviewees after completion of the interviews for comments or corrections. The transcripts were transcribed and translated from Arabic to English by a professional Arabic translator who spoke English as a second language to identify and resolve any issues with the concept or expressions.

  • Was any information collected on whether people had a preference towards only using free mobile apps? Several apps come with a feature of becoming a paid member as that offers additional benefits. It would be interesting to know if similar information was captured in the interviews. If the app is not free, then the tendency to purchase a plan will decrease.

Thanks for your comments! Yes, and this has been addressed in the manuscript on page (5).

Participants have reported that signing up for any mobile app should be a stress-free procedure, especially as it is the first interaction they will have with the application. Therefore, making registration easy, free, and quick in the apps is an important feature that needs to be taken into consideration. Most of the participants in this study had a preference for using free mobile apps. They believed that registering via an external account facilitates the registration of the apps, as external accounts can be used to speed up the registration procedure even more.

Reviewer 2 Report

Please check the sticky note. There is some information missing about trustworthiness of the study. 

Author Response

We greatly appreciate the reviewers’ enthusiasm about manuscript and the quality comments offered by the reviewers. The suggestions offered by the reviewers have been immensely helpful, and we also appreciate your insightful comments on revising the paper. We have included the reviewers’ comments in the following table and responded to them individually, describing the changes we have made. Also, all changes are highlighted and have been incorporated to the manuscript, and the revised manuscript is attached.

Author's Reply to the Review Report (Reviewer 2)

Please check the sticky note. There is some information missing about trustworthiness of the study. 

Thanks for your valuable comments, these comments have been addressed accordingly, please see page (3)

All interviews (n = 21) were conducted face-to-face at the workplace office of each of the interviewees, and the length of the interviews ranged from 20 to 30 mins, with no repeat interviews conducted. As Arabic is the formal language in Saudi Arabia, all of interviews were conducted in Arabic language. Participant's validation was achieved by returning transcripts to interviewees after completion of the interviews for comments or corrections. The transcripts were transcribed and translated from Arabic to English by a professional Arabic translator who spoke English as a second language to identify and resolve any issues with the concept or expressions. We ensured that there was no social desirability bias in this study as there was no prior relationship between the interviewees and interviewer.

The other valuable comments on the sticky note have been addressed and incorporated in the manuscript (Please see the green highlights). “Please see the attachment”

Author Response

We greatly appreciate the reviewers’ enthusiasm about manuscript and the quality comments offered by the reviewers. The suggestions offered by the reviewers have been immensely helpful, and we also appreciate your insightful comments on revising the paper. We have included the reviewers’ comments in the following table and responded to them individually, describing the changes we have made. Also, all changes are highlighted and have been incorporated to the manuscript, and the revised manuscript is attached.

Author's Reply to the Review Report (Reviewer 3)

The original research entitled "Exploring Saudi Individuals' Perspectives and Needs to Design a Hypertension Management Mobile Technology Solution: Qualitative study" evaluated Saudi individuals' perspectives and needs concerning designing a hypertension management Mobile app. The information was obtained through personal surveys in a qualitative format.

The researchers did not report inclusion and exclusion criteria of the subjects surveyed.

Thank you for your insightful comments, the inclusion and exclusion criteria have been addressed (please see page 2 )

Participants were eligible to participate if they had worked experience in hypertension management. Also, the inclusion criteria included patients who have hypertension disease for at least one year and were older than 18 years. All participants meeting these criteria were approached to participate in the study.

 They were recruited through researcher's contacts and a snowballing technique. This is a very important limitation of the study since the sampled population is not defined and its representativeness cannot be established. The sample size to achieve meaningful information was not establish and the sampling was stopped at 21 surveys because no new themes were emerging. In this scenario the power and significance of the obtained information cannot be weighted.

Thanks for your valuable comments, these comments have been addressed and incorporated in the manuscripts under the limitation subheading. Please see the blue highlight in the manuscript on page (10)

The authors chose a qualitative approach to report the information. A more quantitative survey approach with questions with dichotomous, Likert-scale, rank-order, and open-ended response choices could be more explanatory. 

Thanks for your comments and the authors will be working on this on the second phase of this project. This paper presents the first phase of a mixed method project which deals with the design and development of a usable mobile technology solution to self-monitor Hypertension (HTN) and improve health and fitness levels among Saudi adults.  The second phase of the project will be using quantitative approach to test the usability of the app. Therefore, this paper is conducted in qualitative approach to lay the foundation for the whole project. The implication and future research subheading has been created to address these comments.  Please see Page 9

Implications for practice and future research

The current study's findings shed light on understanding the variety of user experiences and expectations of potential future intervention users. These can help designers create mobile health apps that encourage self-monitoring to improve self-management of hypertension. It provides practical insights into the challenges and needs of patients with hypertension, highlighting key areas to be addressed by policymakers and stakeholders to develop and design mobile health apps to improve the management of hypertritons. Future studies evaluating hypertension patients' needs should involve a quantitative approach with large samples of participants for a deeper understanding of the phenomenon and effectively explore mobile health app features that could help improve the management of hypertension and other chronic conditions.

Surprisingly, any data about surveyed expectations of the impact of mHealth-Apps on their own health and quality of life, credibility, usefulness, etc., were published which in opinion of this reviewer are very important.

Thanks for your supportive comments…

The discussion section should be shortened and sharply focus on the research results.

The discussion section has been revised accordingly.

Finally, a limitation section should be added explaining sampling representative and issues, sample size and limitations of the qualitative approach.

Thank you for your comments, a limitation section has been added please see page (10).

Limitations and Strengths 

Limitations of this study include conveniently drawn samples from the sample size. Nevertheless, the sample size was sufficient to identify the requirements and needs of patients and medical experts to design a hypertension mobile app for Saudi Patients. As this research implemented a qualitative study design, we could not make causality assumptions, which is a limitation when interpreting the findings. Additionally, qualitative research is not statistically representative. However, it is much more appropriate for our research that seeks to understand individuals' perspectives and needs in relation to designing a hypertension management Mobile app to be used by hypertension patients to manage their condition better. To the authors' best knowledge, this study is the first of its kind to gain an in-depth understanding of participants' personal needs and suggestions for designing and developing Mobile health apps for patients with hypertension in Saudi Arabia.

Round 2

Reviewer 3 Report

The observations were resolved